# The SGLT2 Inhibitor Canagliflozin Prevents Carcinogenesis in a Mouse Model of Diabetes and Non-Alcoholic Steatohepatitis-Related Hepatocarcinogenesis: Association with SGLT2 Expression in Hepatocellular Carcinoma

**DOI:** 10.3390/ijms20205237

**Published:** 2019-10-22

**Authors:** Teruo Jojima, Sho Wakamatsu, Masato Kase, Toshie Iijima, Yuko Maejima, Kenju Shimomura, Takahiko Kogai, Takuya Tomaru, Isao Usui, Yoshimasa Aso

**Affiliations:** 1Department of Endocrinology and Metabolism, Dokkyo Medical University, Tochigi 321-0293, Japan; jojima@dokkyomed.ac.jp (T.J.); syowaka@dokkyomed.ac.jp (S.W.); k-masato@dokkyomed.ac.jp (M.K.); toshie@dokkyomed.ac.jp (T.I.); ttomaru@dokkyomed.ac.jp (T.T.); isaousui@dokkyomed.ac.jp (I.U.); 2Department of Pharmacology, Fukushima Medical University School of Medicine, Fukushima 960-1295, Japan; maejimay@fmu.ac.jp (Y.M.); shimomur@fmu.ac.jp (K.S.); 3Department of Infection Control and Clinical Laboratory Medicine, Dokkyo Medical University, Tochigi 321-0293, Japan; tkogai@dokkyomed.ac.jp

**Keywords:** non-alcoholic steatohepatitis, non-alcoholic fatty liver disease, hepatocellular carcinoma, carcinogenesis, sodium-glucose co-transporter 2, liver fibrosis, cell cycle

## Abstract

The aim of the present study is to investigate the effects of canagliflozin, a selective sodium-glucose co-transporter 2 (SGLT2) inhibitor, on non-alcoholic steatohepatitis (NASH) and NASH-related hepatocellular carcinoma (HCC) in a mouse model of diabetes and NASH-HCC. First, mice aged five weeks were divided into two groups (vehicle group and canagliflozin group) and were treated for three weeks. Then, mice aged five weeks were divided into three groups of nine animals each: the vehicle group, early canagliflozin group (treated from five to nine weeks), and continuous canagliflozin group (treated from five to 16 weeks). Canagliflozin was administered at a dose of 30 mg/kg in these experiments. In addition, the in vitro effects of canagliflozin were investigated using HepG2 cells, a human HCC cell line. At the age of eight or 16 weeks, the histological non-alcoholic fatty liver disease activity score was lower in the canagliflozin-treated mice than in vehicle-treated mice. There were significantly fewer hepatic tumors in the continuous canagliflozin group than in the vehicle group. Immunohistochemistry showed significantly fewer glutamine synthetase-positive nodules in the continuous canagliflozin group than in the vehicle group. Expression of α-fetoprotein mRNA, a marker of HCC, was downregulated in the continuous canagliflozin group when compared with the vehicle group. At 16 weeks, there was diffuse SGLT1 expression in the hepatic lobules and strong expression by hepatocytes in the vehicle group, while SGLT2 expression was stronger in liver tumors than in the lobules. In the in vitro study, canagliflozin (10 μM) suppressed the proliferation of HepG2 cells. Flow cytometry showed that canagliflozin reduced the percentage of HepG2 cells in the G2/M phase due to arrest in the G1 phase along with decreased expression of cyclin D and Cdk4 proteins, while it increased the percentage of cells in the G0/1 phase. Canagliflozin also induced apoptosis of HepG2 cells via activation of caspase 3. In this mouse model of diabetes and NASH/HCC, canagliflozin showed anti-steatotic and anti-inflammatory effects that attenuated the development of NASH and prevented the progression of NASH to HCC, partly due to the induction of cell cycle arrest and/or apoptosis as well as the reduction of tumor growth through the direct inhibition of SGLT2 in tumor cells.

## 1. Introduction

Hepatocellular carcinoma (HCC) is the sixth most common cancer overall and the third leading cause of cancer-related death worldwide [1]. Non-alcoholic steatohepatitis (NASH) is characterized by hepatocellular accumulation of lipids (steatosis), lobular inflammation, hepatocellular ballooning, and fibrosis [2]. NASH is associated with an increased risk of cirrhosis and HCC [3], and has emerged as a major risk factor for HCC. Patients with type 2 diabetes are susceptible to developing severe NASH, and also have a higher risk of NASH progressing to cirrhosis and/or HCC when compared with non-diabetic persons [4].

Several studies have examined the impact of sodium-glucose co-transporter 2 (SGLT2) inhibitors on the occurrence of non-alcoholic fatty liver disease (NAFLD) and/or NASH in rodent models and humans [5,6,7,8,9]. We previously reported that the SGLT2 inhibitor empagliflozin showed anti-steatotic and anti-inflammatory effects [5], which suppressed the development of NASH in a mouse model of diabetes and NASH/cirrhosis/HCC (the STAM mouse) [10]. This mouse model progresses from diabetes with NAFLD to NASH by eight weeks of age and HCC develops at 16 weeks [10]. We recently reported that the SGLT2 inhibitor dapagliflozin improves hepatic steatosis in patients with type 2 diabetes and NAFLD, and attenuates liver fibrosis in patients with significant fibrosis [11], suggesting that SGLT2 inhibitors have an anti-fibrotic effect on human NASH. It is therefore possible that SGLT2 inhibitors could prevent NASH-related hepatocarcinogenesis in the setting of diabetes combined with NAFLD. However, no studies have examined whether SGLT2 inhibitors can prevent the progression of NASH to hepatocarcinogenesis.

Canagliflozin is an orally administered SGLT2 inhibitor used in the treatment of patients with type 2 diabetes. By inhibiting the transporter protein SGLT2 in the kidneys, canagliflozin reduces renal glucose reabsorption, thereby increasing urinary glucose excretion and reducing blood glucose levels [12]. A previous study has demonstrated that canagliflozin lowers the renal threshold for glucose excretion, increases urinary glucose excretion, reduces body weight, increases fatty acid oxidation, and reduces de novo lipogenesis in rodent models of type 2 diabetes [13].

Accordingly, we investigated the effect of the SGLT2 inhibitor canagliflozin on steatohepatitis and NASH-related hepatocarcinogenesis in the STAM mouse model of diabetes and NASH/cirrhosis/HCC. We also evaluated the in vitro effect of canagliflozin on the proliferation of an HCC cell line (HepG2 cells) and the mechanisms involved.

## 2. Results

### 2.1. Effect of Canagliflozin on the NAFLD Histological Activity Score (NAS) and Hepatic Fibrosis

As shown in Figure 1, mice aged five weeks were divided into two groups (a vehicle group and a canagliflozin group). The liver/body weight ratio was lower in the canagliflozin group than in the vehicle group. Plasma levels of glucose and triglycerides were significantly lower in the canagliflozin group when compared with the vehicle group (Appendix A).

In mice from the vehicle group, examination of H-E-stained liver sections revealed fatty degeneration, inflammatory cell infiltration, and hepatocyte ballooning, with these changes being predominantly around the central veins (Appendix A). The NAFLD activity score (NAS) was significantly lower in the canagliflozin group when compared with the vehicle group (Appendix A). Expression of the suppressor of cytokine signaling (SOCS) 3 gene was decreased in the canagliflozin group (Appendix A). Although Sirius red staining showed no significant difference in the area of collagen deposition between the canagliflozin and vehicle groups (Appendix A), expression of type 3 collagen mRNA was significantly lower in the canagliflozin group than in the vehicle group (Appendix A).

### 2.2. Effect of Canagliflozin on Body Weight, Liver Weight, and Biochemical Parameters

As shown in Figure 1, mice aged five weeks were divided into three groups of nine animals each: the vehicle group, the early canagliflozin group (receiving canagliflozin at 30 mg/kg daily from weeks 5 to 9), and the continuous canagliflozin group (receiving canagliflozin at 30 mg/kg daily from weeks 5 to 16). The liver/ body weight ratio was lower in the continuous canagliflozin group than in the vehicle group or the early canagliflozin group (Table 1). In addition, the plasma levels of glucose and ALT were significantly lower in the continuous canagliflozin group when compared with the vehicle group or the early canagliflozin group (Table 1).

#### 2.2.1. Effect of Early Canagliflozin Administration (5–9 Weeks) or Continuous Canagliflozin Administration (5–16 Weeks) on the NAS

In the vehicle group, examination of H-E-stained liver sections revealed fatty degeneration, inflammatory cell infiltration, and hepatocyte ballooning, predominantly around the central veins (Figure 2a). The continuous canagliflozin group had a significantly lower NAS score than either the vehicle group or the early canagliflozin group (Figure 2b). Scores for each NAS component in all groups are displayed in Figure 2c.

We also investigated whether canagliflozin influenced hepatic lipid metabolism. We found that expression of mRNA for FAS, a gene involved in fatty acid production (lipogenesis), was significantly lower in the continuous canagliflozin group than in the vehicle group or early canagliflozin group (Figure 2d). Expression of mRNA for ACC1, another gene involved in lipogenesis, was also lower in the continuous canagliflozin group than in the other groups, but the difference was not significant. On the other hand, there were no significant differences among the groups with regard to the expression of mRNAs for genes related to β-oxidation such as PPAR-α and ACOX-1 (data not shown). 

#### 2.2.2. Effect of Canagliflozin on Hepatic Fibrosis

We next investigated whether canagliflozin prevented the progression of hepatic fibrosis, which is the advanced stage of NASH. First, liver fibrosis was assessed by Sirius red staining (Figure 3a), revealing that the area of collagen deposition was significantly smaller in the early canagliflozin group relative to the vehicle group (Figure 3b). In addition, expression of type 3 collagen mRNA was significantly lower in the early canagliflozin group than in the vehicle group (Figure 3c). 

#### 2.2.3. Canagliflozin Inhibits Hepatic Tumorigenesis

At 16 weeks, liver tumors were found in some of the STAM mice from each group (Figure 4a). There were significantly fewer tumors in the continuous canagliflozin group than in the vehicle group (Figure 4b), although there was no significant difference of tumor size among the three groups (Figure 4c). Examination of H-E-stained liver sections showed that the liver tumors were the foci of abnormal cells with basophilic cytoplasm and hyperchromic nuclei, suggesting the development of HCC (Figure 4d).

#### 2.2.4. Canagliflozin Inhibits Progression of NASH to Hepatocarcinogenesis

Next, to investigate whether canagliflozin prevented the progression of NASH to HCC by inhibiting hepatocarcinogenesis, immunohistochemical staining was performed for glutamine synthetase (GS), which is a marker of HCC. Positive signals for GS (characteristic of HCC) were detected in the tumor nodules (Figure 5a). GS-positive nodules were found in four out of five mice from the vehicle group (treated with the vehicle from five to 16 weeks), three out eight mice in the continuous canagliflozin group (treated with canagliflozin from five to 16 weeks), four out six mice in the early canagliflozin group (treated with canagliflozin from five to eight weeks and sacrificed at 16 weeks), and one out three mice in the canagliflozin group (treated with canagliflozin from five to eight weeks and sacrificed at eight weeks). GS-positive nodules were not detected in the mice from the vehicle group sacrificed at eight weeks. There were fewer GS-positive nodules in the continuous canagliflozin group when compared with mice from the vehicle group sacrificed at 16 weeks (Figure 5b).

Expression of mRNA for AFP, an oncofetal protein that is used as a tumor marker, was significantly lower in the continuous canagliflozin group than in the vehicle group (Figure 5c). Kaplan–Meier survival curves for STAM mice receiving the indicated treatments are depicted in Figure 5d. The continuous canagliflozin group showed significantly better survival when compared with the vehicle group (*p* < 0.001).

#### 2.2.5. Hepatic and Tumor SGLT1/SGLT2 Expression in the Vehicle Group at Eight and 16 Weeks

In the vehicle group, immunohistochemistry for SGLT1 and SGLT2 using specific antibodies revealed diffuse SGLT1 expression in the hepatic lobules and strong expression by hepatocytes (Figure 6a). Hepatic expression of SGLT1 was stronger in vehicle mice aged 16 weeks than in those aged eight weeks (Figure 6b). On the other hand, SGLT2 expression was seen in the hepatic lobules of vehicle mice aged 16 weeks (Figure 6d), but not in mice aged eight weeks (Figure 6c). In particular, there was strong SGLT2 expression in the liver tumors of mice aged 16 weeks (Figure 6d).

### 2.3. Effects of Canagliflozin on Real-Time Proliferation of HepG2 Cells Evaluated by the xCELLigence DP System

We examined SGLT1 and SGLT2 protein expression in HepG2 cells and found obvious expressions of both SGLT1 and SGL2 (Appendix A). Then, we investigated the effect of canagliflozin on proliferation of HepG2 cells by using a proliferation assay (Figure 7a). Treatment with 10 μM canagliflozin for 144 h significantly inhibited the proliferation of HepG2 cells relative to treatment with the vehicle or with 100 nM canagliflozin (Figure 7b). We subsequently investigated the effect of 10 μM canagliflozin on cell cycle progression by flow cytometry, revealing that treatment for 24 h significantly increased the population of cells in G0/G1 phase and decreased the population in the G2/M phase when compared with the control (Figure 7c).

Although not statistically significant, the percentage of S phase population cells tended to be lower after treatment with 10 μM of canagliflozin than the control (22.6 ± 4.1 vs. 29.4 ± 6.5%, *p* = 0.0992).

#### Effects of Canagliflozin on Cell Cycle-Related Proteins and Apoptosis

Western blot analysis showed that treatment with 10 μM canagliflozin significantly reduced cyclin D protein expression when compared with the other three groups (Figure 8a,b). In addition, cdk4 protein expression was significantly lower after treatment with 10 μM canagliflozin than after treatment with 100 nM canagliflozin (Figure 8a,b). qRT-PCR also demonstrated that the gene expression of CCND1 was reduced in treatment with 10 μM canagliflozin when compared with the control (Appendix A). Furthermore, we evaluated whether canagliflozin induced apoptosis of HepG2 cells by using flow cytometry with annexin V antibody. We demonstrated that treatment with 10 μM canagliflozin significantly increased the percentage of annexin V-positive cells when compared with the control or treatment with 100 nM canagliflozin (Figure 8d). Treatment with 10 μM canagliflozin also augmented the expression of cleaved caspase 3 protein (Figure 8a,c).

## 3. Discussion

The present study demonstrated that treatment with canagliflozin prevented the development of NASH in a mouse model of diabetes with NASH/HCC. The NAS was significantly lower in animals treated with canagliflozin from five to eight weeks of age when compared with the vehicle group (Appendix A). This finding was in agreement with the results of our previous study, which showed that treatment with empagliflozin (6–9 weeks of age) ameliorated hepatic steatosis, inflammation, and fibrosis in the same mouse model [5]. In addition, a prospective study demonstrated that canagliflozin treatment improved liver histology in patients with NAFLD and type 2 diabetes [14]. One possible explanation for such an improvement in hepatic steatosis and inflammation by canagliflozin is the inhibition of de novo lipogenesis, since the present study showed that expression of FAS, a gene involved in lipogenesis [15], was significantly lower in the canagliflozin group than in the vehicle group. FAS is a key enzyme for the hepatic biosynthesis of fatty acids, and is thought to be a determinant of the maximal capacity of the liver to synthesize fatty acids by de novo lipogenesis because it catalyzes the last step in this biosynthetic pathway [16]. Thus, improvement of NASH by canagliflozin may be associated with a decrease of fatty acid production (de novo lipogenesis) rather than an increase of fatty acid oxidation.

In this study, we demonstrated that continuous administration of canagliflozin prevented the occurrence of HCC by inhibiting tumorigenesis because there were significantly fewer tumors in the continuous canagliflozin group than in the vehicle group. Shiba et al. previously reported that canagliflozin attenuates the development of HCC in melanocortin 4 receptor-deficient mice fed a Western diet, a mouse model of human HCC, with the number of liver tumors being significantly reduced by canagliflozin treatment when compared with the placebo [17]. However, they did not perform histological or immunohistochemical analysis to confirm that the tumors occurred via the process of hepatocarcinogenesis. Another study showed that canagliflozin reduced the burden of subcutaneous HepG2-derived xenograft tumors in BALB/c nude mice, but did not confirm the prevention of the progression of NASH to carcinogenesis [18].

Accordingly, this study provided the first evidence that canagliflozin inhibits the progression of NASH to hepatocarcinogenesis, since the number of GS-positive tumors was significantly smaller in the continuous canagliflozin group than in the vehicle group. GS is a marker of HCC [19], so we used immunohistochemistry for GS to identify the liver tumors. We detected positive signals for GS in the tumor nodules, supporting a diagnosis of HCC. In the mammalian liver, GS catalyzes the synthesis of glutamine from glutamate and ammonia, and thus provides a source of energy for tumor cells. Immunohistochemical analysis has shown a stepwise increase of GS expression from precancerous lesions to early and advanced HCC [20]. GS-positive malignant cells are derived from GS-positive hepatocytes, suggesting that GS is a specific marker for tracing cell lineage during the process of hepatocarcinogenesis. We also found that hepatic expression of mRNA for AFP (an oncofetal protein and tumor marker) was significantly lower in the continuous canagliflozin group than in the vehicle group. AFP is known to be a marker of hepatic progenitor cells [21]. Taken together, these findings indicate that canagliflozin inhibited hepatocarcinogenesis in this mouse model of diabetes/NASH/HCC.

Our in vitro study demonstrated that treatment with 10 μM canagliflozin significantly inhibited the proliferation of HepG2 cells relative to the control or treatment with 100 nM canagliflozin, suggesting that canagliflozin had an antiproliferative effect on this human HCC cell line, although the mechanisms involved are still unclear. Cell cycle analysis showed that canagliflozin treatment reduced the cell population in the G2/M phase, accompanied by a decrease of cyclin D and Cdk4 protein expression when compared with the control, while it increased the G0/1 phase population. Therefore, it seems that canagliflozin inhibited HCC proliferation by inducing G2/M arrest (Figure 9). Our results are in agreement with a previous report that canagliflozin treatment caused downregulation of cyclin d1 and 2 in a human HCC cell line [18]. Another possibility is that canagliflozin induces apoptosis of HCC cells. In fact, treatment with 10 μM canagliflozin significantly increased the percentage of annexin V-positive cells when compared with the control or treatment with 100 nM canagliflozin. Moreover, cleavage of caspase 3 was enhanced, suggesting that canagliflozin promoted apoptosis of HepG2 cells by activating caspase-3 (Figure 9).

It is also possible that the anti-steatotic and anti-inflammatory effects of canagliflozin prevented the development of hepatocarcinogenesis and HCC in this diabetes/NASH/HCC model. In the present study, the NAS score was significantly lower in the continuous canagliflozin group when compared with the vehicle group or the early canagliflozin group. Additionally, in people with NAFLD, a prospective study demonstrated that the severity of liver fibrosis is an independent predictor of liver-related mortality including death from HCC and cirrhosis [22]. Sirius red staining showed that compared with the vehicle group, collagen deposition was significantly reduced in the early canagliflozin group, but not in the continuous canagliflozin group, suggesting that the mechanisms by which canagliflozin ameliorated NASH or prevented liver cancer may be not the same. We speculate that canagliflozin may have a direct inhibitory effect on the occurrence of HCC.

Tumors have an increased demand for glucose that is utilized to synthesize ATP by aerobic glycolysis [23]. Uptake of glucose into cancer cells is promoted by facilitative diffusion mediated via GLUTs, and the overexpression of GLUT1 is well documented in various cancers [23]. Functional SGLT2 expression has been demonstrated in pancreatic and prostate carcinoma, while inhibition of SGLT2 by canagliflozin blocks glucose uptake to reduce tumor growth and improve survival in a xenograft model of pancreatic cancer [24]. We found that both SGLT1 and SGLT2 were prominently expressed by HepG2 cells, in agreement with other reports [18,25]. We also demonstrated for the first time that in this mouse model of diabetes with NASH/cirrhosis/HCC at 16 weeks, there was diffuse SGLT1 expression in the hepatic lobules and strong expression by hepatocytes in the vehicle group, while SGLT2 expression was stronger in liver tumors than in the lobules. If HCC expresses SGLT2, glucose uptake by cancer cells could be reduced by SGLT2 inhibitors, thus inhibiting aerobic glycolysis and suppressing tumor growth (Figure 9). Since we found that continuous administration of canagliflozin, but not early treatment, prevented the occurrence of HCC by inhibiting tumorigenesis, canagliflozin may exert antiproliferative effect through direct inhibition of SGLT2 in tumor cells.

The present study had some limitations. There was a significant difference of the plasma glucose level between the canagliflozin and vehicle groups after treatment, so the possibility that canagliflozin partly prevented NASH and NASH-related HCC by its glucose-lowering effect could not be excluded. Another limitation is that the present mouse model lacks obesity and insulin resistance, which are associated with the pathogenesis of NASH in humans. The third limitation is the dosage of canagliflozin used in our in vivo studies, since canagliflozin 30 mg/kg/day is a super pharmacological dosage.

In conclusion, canagliflozin showed anti-steatotic and anti-inflammatory effects that attenuated the development of NASH in the present mouse model of diabetes/NASH/HCC, and prevented the progression of NASH to HCC, partly due to the induction of cell cycle arrest and/or apoptosis, or the reduction of tumor growth through the direct inhibition of SGLT2 in tumor cells.

## 4. Materials and Methods

### 4.1. Animal Model and Induction of NASH

The mouse model of NASH (STAM mouse) was generated according to the protocol of Fujii et al. [10]. Pathogen free 14-day pregnant C57BL/6 mice were purchased from CLEA Japan (Tokyo, Japan). Male pups (two days old) received a subcutaneous injection of a low dose of streptozotocin (200 μg) and then were fed a high-fat diet (HFD32; CLEA-Japan, Tokyo, Japan) ad libitum from four weeks after birth. This mouse model progresses from NAFLD to NASH by eight weeks of age, hepatic fibrosis by 12 weeks, and HCC (tumor protrusion) develops at 16 weeks [10]. Canagliflozin was provided by Mitsubishi Tanabe Pharma Corporation (Osaka, Japan). The dose of canagliflozin was set at 30 mg/kg in all experiments because it has been shown to be an effective pharmacological dose in animal studies [26]. All experiments were approved by the animal care and use committee of Dokkyo Medical University (SLMN 034-1603-3). All animal methods were performed in accordance with the relevant guidelines and regulations prepared by the National Academy of Sciences and published by the National Institute of Health.

#### 4.1.1. Study 1

We investigated whether canagliflozin could prevent the progression of NAFLD to NASH. As shown in Figure 1, mice aged five weeks were divided into two groups (a vehicle group and a canagliflozin group). All animals were provided with food and water ad libitum, with canagliflozin being administered at 30 mg/kg once daily by oral gavage for three weeks in the canagliflozin group. At the end of the study period, the mice were sacrificed for the collection of liver tissue samples and blood was obtained via cardiac puncture at the time of death. Blood samples were centrifuged and serum was stored at −80 °C until analysis. The fasting blood glucose level (16 h fast) was measured with a Glutest Neo Sensor (Sanwa Chemical, Shizuoka, Japan). Serum levels of alanine aminotransferase (ALT), triglycerides (TG), and glycated albumin (GA) were measured with an automated analyzer (JEOL Ltd., Tokyo, Japan). Serum insulin was measured with an enzyme immunoassay (Shibayagi Co., Shibukawa, Japan).

#### 4.1.2. Study 2

We also investigated the inhibitory effect of canagliflozin on HCC in this diabetes/NASH/HCC model. As shown in Figure 1, mice aged five weeks were divided into three groups of nine animals each: the vehicle group, the early canagliflozin group (receiving canagliflozin at 30 mg/kg daily from weeks five to 9), and the continuous canagliflozin group (receiving canagliflozin at 30 mg/kg daily from weeks five to 16). All animals were provided with food and water ad libitum, with canagliflozin being administered at 30 mg/kg once daily by oral gavage in the canagliflozin groups. At 16 weeks, the mice were sacrificed for collection of liver tissue samples and blood was obtained via cardiac puncture at the time of death. Blood samples were centrifuged and serum was stored at −80 °C until analysis.

Samples from the left lobe of the liver were embedded in tissue Tek O.C.T. compound (Sakura Finetek, Tokyo, Japan), snap frozen in liquid nitrogen, and stored at −80 °C. Sections were cut (5 μm), dried in air, and fixed in acetone. For hematoxylin and eosin (H-E) staining, liver sections were prefixed in Bouin’s solution and then stained with Lillie–Mayer hematoxylin (Muto Pure Chemicals, Tokyo, Japan). Based on observation of the H-E-stained sections, the NAFLD histological activity score (NAS) was calculated from the severity of steatosis, lobular inflammation, and hepatocyte ballooning according to the criteria of Kleiner et al. [27].

To assess fibrosis, liver sections were stained with Sirius red, and the Sirius red-positive areas in five non-overlapping fields were measured with DP manager/controller and WIN Roof Ver.5.8.1 software (Mitani Co., Tokyo, Japan).

Immunohistochemical staining was performed with anti-glutamine synthetase (GS) antibody (Chemicon International, Temeculla, CA, USA), anti-SGLT1 antibody (Abcam, Cambridge Science Park, Cambridge, UK), and anti-SGLT2 antibody (Santa Cruz Biotechnology, Texas, USA). For quantitative analysis of the anti-GS-positive area, bright images of the region around the central vein were captured at 200x magnification using a digital camera (BX53 Olympus Japan, Tokyo, Japan). Then, the anti-GS-positive areas in three to five non-overlapping fields were measured with DP manager/controller and WIN Roof Ver.5.8.1 software.

### 4.2. Cell Culture and Reagents

A human hepatocellular carcinoma cell line, HepG2, was obtained from RIKEN BRC (Ibaraki, Japan). Monolayer cultures were maintained in Dulbecco’s modified Eagle’s medium supplemented with 10% fetal bovine serum (both from Invitrogen, Waltham, MA) and 1% penicillin/streptomycin in an incubator at 37 °C under a humidified atmosphere with 5% CO_2_. Cells at passages five to 12 were used for these experiments. Canagliflozin was provided by Mitsubishi Tanabe Pharma Corporation (Osaka, Japan). THP-1 cells (a human monocytic cell line) and human umbilical vein endothelial cells (HUVEC) were obtained from Clonetics Co. (San Diego, CA 92123, USA).

### 4.3. Measurement of Cell Proliferation

The xCELLigence DP system was used for real-time assessment of cell proliferation, migration, invasion, and viability as previously described [28]. This system employs special single-use culture plates with gold electrode arrays covering 80% of the 5.0-mm area at the bottom of each individual well. An increase of cellular contact with the electrodes increases impedance across the array, and impedance can be measured by the DP system in arbitrary cell index units (E-plates) to assess cell growth and viability. HepG2 cells (8 × 10^4^) were added to each well and allowed to settle for 30 min before any treatment. Then, the cells were incubated with 100 nM canagliflozin (TA-7284; Mitsubishi Tanabe Pharma Corporation, Saitama, Japan), 1 μM canagliflozin, or 10 μM canagliflozin and the cell index was measured every 15 min for six days (144 h).

Canagliflozin inhibited glucose transport by the cells at a concentration of 10 μM, which corresponds to the serum concentration at clinical doses [13].

### 4.4. Western Blotting

After treatment, HepG2 cells were lysed in cell lysis buffer (Cell Signaling Technology, Beverly, MA, USA) containing 1 mmol/l phenylmethylsulfonyl fluoride. As previously described [26], the protein concentration in each sample was measured with a Bio-Rad detergent-compatible protein assay (Bio-Rad laboratory, Tokyo, Japan). Samples containing 10 μg of protein were resolved by electrophoresis on 12% sodium dodecyl sulfate-polyacrylamide gel and transferred to a polyvinylidene difluoride membrane (Bio-Rad), followed by incubation with primary antibodies targeting cyclin D, cyclin-dependent protein 4(CDK4), and β-actin [28]. After incubation with the secondary antibody (horseradish peroxidase-conjugated sheep anti-rabbit IgG, 1:20,000), reaction products were detected with an Amersham ECL Plus system and an enzyme-linked chemiluminescence detection kit (Amersham Biosciences, Piscataway, NJ, USA), and the band density was quantified using a LumiVision Analyzer (Aisin, Kariya, Japan). The following primary antibodies were used: anti-cyclin D, anti-p21, anti-p53, anti-p70S6, anti-cleaved caspase3, anti-caspase3, and anti-β-actin (all from Cell Signaling Technology, Beverly, MA, USA), anti-Cdk4, anti-sodium glucose co-transporter 1, and anti-sodium glucose co-transporter 2 (al from Abcam, Cambridge Science Park, Cambridge, UK).

### 4.5. Quantitative Real-Time RT-PCR

Total RNA was extracted from liver tissue samples and HepG2 cells by using an SV Total RNA Isolation System (Promega, Madison, WI, USA) and reverse transcription was done with SuperScript III First Strand Synthesis Super mix (Invitrogen, Carlsbad, CA, USA) to obtain cDNA [5,28]. Then, real-time quantitative RT-PCR was performed using FastStart SYBR Green Master Mix (Roche Applied Science, Mannheim, Germany) and the following probes: type 1 collagen (NM-007742.3), type 3 collagen (NM-009930.2), ACC1 (NM-133360.2), FAS (NM-134383.2), PPAR-α (NM-001113418.1), ACOX-1 (NM-015729.3), SOCS3 (NM-007707.3), CCND1 (Rn_00432359_m1), and AFP (Mm00431715-m1).

Expression of mRNAs for SOCS3, type 1 collagen, type 3 collagen, FAS, ACC-1, PPAR-α, ACOX-1, SOCS3, CCND1, or AFP was normalized for the expression of GAPDH mRNA [5,26].

### 4.6. Cell Cycle Analysis

To synchronize the cell cycle, HepG2 cells were subjected to serum deprivation for 6 h before treatment with canagliflozin and then resupplied with serum. The cells were subsequently incubated with 10 μM canagliflozin for 24 h and harvested by trypsinization, after which 1 × 10^6^ cells were stained with 100 μg/mL propidium iodide in the dark at 4 °C. The fraction of cells in each phase of the cell cycle (G0/G1, S, and G2/M) was determined by flow cytometry using a BD FACStar with ModiFit software (Becton Dickinson Japan, Tokyo, Japan). A canagliflozin concentration of 10 μM corresponds to the serum concentration at a clinically effective dose [13].

### 4.7. Flow Cytometric Analysis

Apoptosis of HepG2 cells was detected by annexin V staining. HepG2 cells were incubated with 20 μL of fluorescein isothiocyanate (FITC)-conjugated Annexin V (Bender Med Systems, Vienna, Austria) in binding buffer for 15 min in the dark at room temperature. After propidium iodide solution was added, the cells were resuspended in 2 mL of phosphate-buffered saline (PBS). A FACS Calibur (Becton Dickinson Japan, Tokyo, Japan) and Cell Quest Pro software (Version 5.1, Becton Dickinson Japan, Tokyo, Japan) were used for flow cytometry. 

### 4.8. Statistical Analysis

Results are presented as the mean ± standard deviation or as the median and interquartile range. Differences between groups were analyzed by the unpaired *t*-test for parametric data or with the Mann–Whitney U test for nonparametric data. Cell cycle data were analyzed by repeated ANOVA (parametric) or the Kruskal–Wallis test (nonparametric). Differences of prevalence between groups were assessed by the chi-square test. In all analyses, a *p* value < 0.05 was accepted as indicating statistical significance.

## Figures and Tables

**Figure 1 ijms-20-05237-f001:**
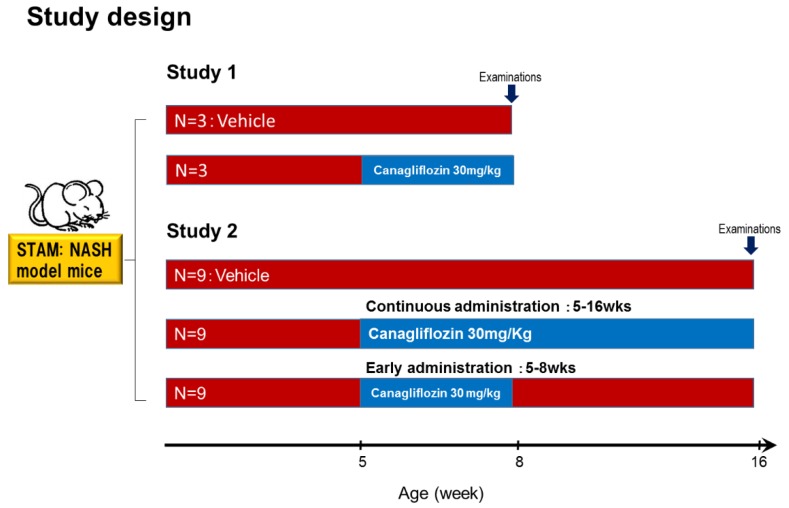
Study design. NASH mice aged five weeks were divided into the vehicle group or canagliflozin (30 mg/kg) group. After grouping, canagliflozin was administered once daily by oral gavage for three weeks. Study 2. NASH-HCC mice aged 5 weeks were divided into three groups of nine animals: vehicle; canagliflozin early administration 30 mg/kg (5 to 9 W); and canagliflozin continuous administration 30 mg/kg (5 to 16 W). After 16 week-treatment, the mice were sacrificed for the collection of liver tissue samples and blood was obtained via cardiac puncture just after death. After grouping, canagliflozin was administered once daily by oral gavage for four and 11 weeks.

**Figure 2 ijms-20-05237-f002:**
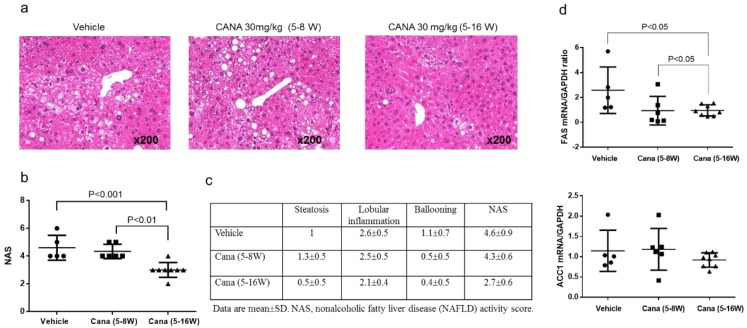
Canagliflozin continuous treatment attenuates steatohepatitis and de novo lipogenesis in the liver in STAM mice. (**a**,**b**) Representative microphotographs of liver sections stained with hematoxylin eosin and NAFLD activity score (NAS) in the three groups. Original magnification, ×200. Data are mean ± SD. (**c**) The scores of each component of NAS. (**d**) mRNA expression of genes involved in lipogenesis.

**Figure 3 ijms-20-05237-f003:**
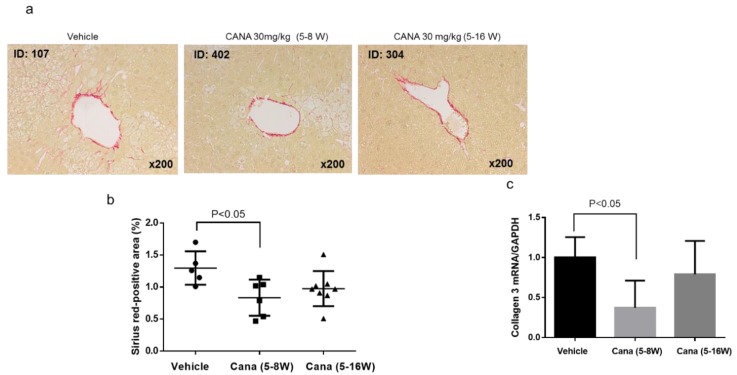
Representative microphotographs of liver sections stained with Sirius red in the liver sections (**a**) and percentage in area of positive staining for Sirius red in the three groups (**b**). (**c**) mRNA expression of collagen 3.

**Figure 4 ijms-20-05237-f004:**
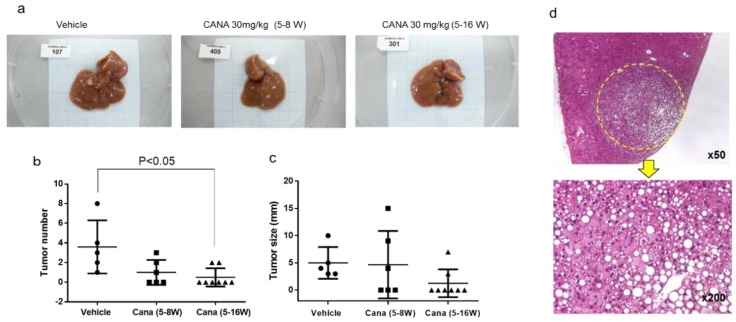
Canagliflozin inhibited liver tumorigenesis. (**a**) Representative images of the whole of liver in the three groups. (**b,c**) The number and size of liver tumors in the three groups. (**d**) Histological findings of H-E showing that the tumor was HCC.

**Figure 5 ijms-20-05237-f005:**
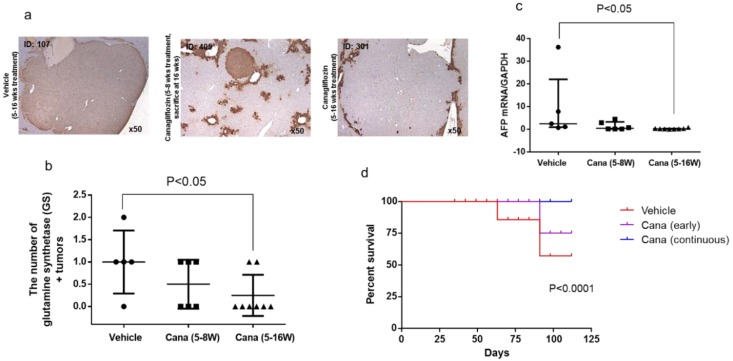
Canagliflozin inhibited progression of hepatic carcinogenesis from NASH. (**a,b**) Representative microphotographs of immunohistochemical staining for glutamine synthetase (GS) in liver sections and percentage in area of positive immunostaining for GS in the three groups. (**c**) mRNA expression of α-fetoprotein in the liver of the three groups. (**d**) Survival curves of STAM mice treated with the three treatments.

**Figure 6 ijms-20-05237-f006:**
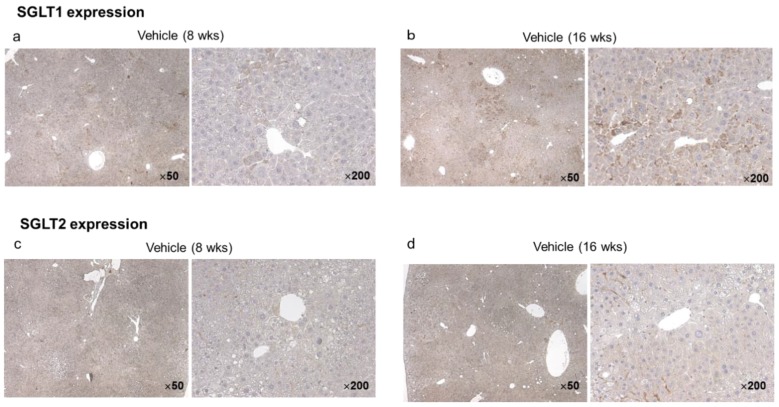
Representative microphotographs of immunohistochemical staining for SGLT1 and SGLT2 in the liver and tumors of vehicle mice ages eight and 16 weeks, respectively. SGLT1 was diffusely expressed in hepatic lobules, and highly expressed in hepatocytes (**a,b**). The expression of SGLT1 in the liver was expressed more intensely in mice aged 16 weeks than in those aged eight weeks (**b**). On the other hand, SGLT2 was expressed in hepatic lobules of mice aged 16 weeks (**d**), but not in those of mice aged eight weeks (**c**). In particular SGLT2 was highly expressed in hepatic tumors of mice aged 16 weeks (**d**).

**Figure 7 ijms-20-05237-f007:**
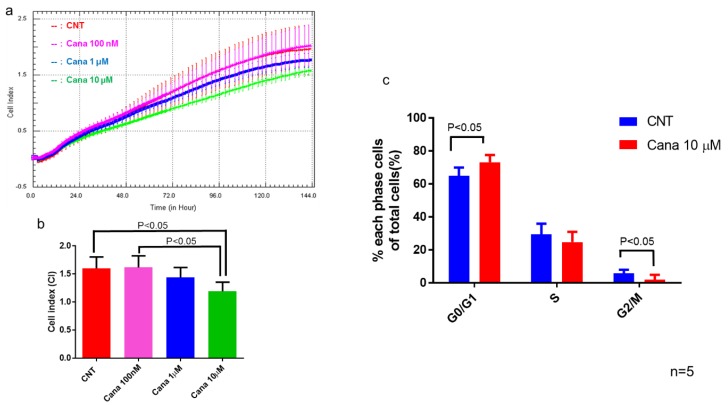
Canagliflozin at a concentration of 10 μM attenuated cell proliferation of HepG2 cells and induce cell cycle arrest. (**a**) Real-time analysis of cell proliferation of HepG2 cells in a dose-dependent manner. (**b**) Treatment with 10 μM canagliflozin inhibited HepG2 cell proliferation when compared with the control or 100 nM canagliflozin. (**c**) Canagliflozin inhibits G2/M cell cycle progression of HepG2 cells.

**Figure 8 ijms-20-05237-f008:**
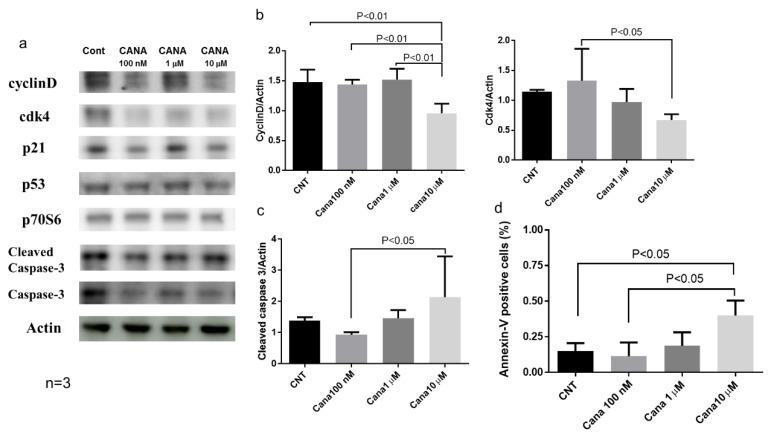
Canagliflozin attenuated proliferation of HCC by inducing cell cycle arrest and/or apoptosis in HepG2 cells. (**a**) Western blot analysis of cell-cycle regulated proteins and cleaved caspase-3 in HepG2 cells. (**b**) The protein expression of both cyclin D and CDK4 significantly reduced in treatment with 10 μM canagliflozin. (**c**) Treatment with 10 μM canagliflozin augmented the protein expression of cleaved caspase 3. (**d**) 10 μM canagliflozin significantly increased the percentage of annexin V-positive cells.

**Figure 9 ijms-20-05237-f009:**
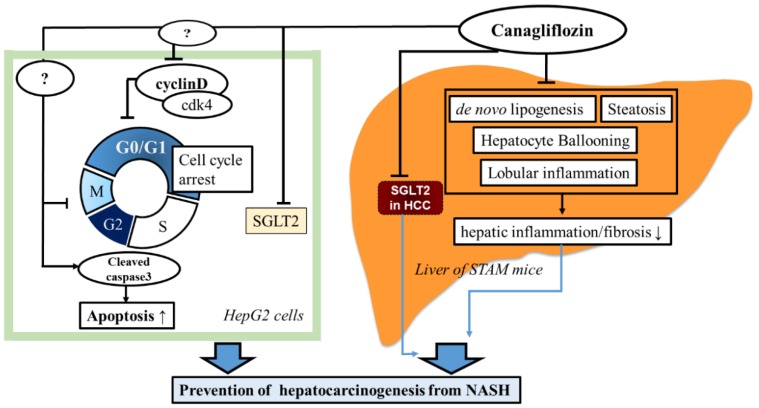
Possible mechanisms responsible for the inhibitory effects of canagliflozin on the progression of hepatocarinogenesis from NASH or NAFLD.

**Table 1 ijms-20-05237-t001:** Body weight and biochemical parameters in NASH mice under diabetic background on day 77 after treatment.

	Vehicle (*n* = 5)	Cana 30 mg/kg (5–8 weeks) (*n* = 6)	Cana 30 mg/kg (5–16 weeks) (*n* = 8)
Body weight (g) Day 0	19.0 ± 1.0	18.7 ± 0.9	18.6 ± 1.0
Body weight (g) Day 77	25.1 ± 1.7	21.5 ± 4.0	25.0 ± 3.5
Liver-to-weight ratio (mg/g)	7.84 ± 1.47	8.77 ± 1.72	6.08 ± 1.35 *
Plasma glucose (mg/dL)	603 ± 91	542 ± 61	310 ± 81 †
ALT (U/L)	74 (48, 82)	86 (43, 226.5)	34 (26.3, 37.8) ‡
Triglyceride (mg/dL)	687.6 ± 432.4	730.3 ± 392.3	514.9 ± 395.1
Serum insulin (pg/mL)	57.9 (39, 443.5)	254 (99, 518.5)	409 (198.8, 1187)

Data are mean ± SD or the median and inter-quartile ranges. * *p* < 0.01 vs. Cana initial (5–8 weeks); † *p* < 0.001 vs. vehicle and vs. Cana initial (5–8 weeks); ‡ *p* < 0.01 vs. vehicle and vs. Cana initial (5–8 weeks), Cana canagliflozin; ALT, alanine aminotransferase.

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
