# Peer review of "The SGLT2 Inhibitor Canagliflozin Prevents Carcinogenesis in a Mouse Model of Diabetes and Non-Alcoholic Steatohepatitis-Related Hepatocarcinogenesis: Association with SGLT2 Expression in Hepatocellular Carcinoma"

_ijms, 2019, doi:10.3390/ijms20205237_

Round 1
Reviewer 1 Report
This paper examines the effect of canagliflozin on NASH/induced HCC in mice.
In Introduction, authors should provide references for efficacy, specificity and pharmacological effects of canagliflozin as an inhibitor of glucose-cotransporter. In Fig. 2, did authors analyze genes of glycolysis pathway, i.e. conversion of glucose to fatty acids ? Fig. 6, the brown staining of SGLT1 and 2 is diffused, appears to be cytosolic, should the staining be on plasma membrane of the hepatocyytes ? Negative control should be shown to exclude false positives. Western blot of these proteins would be better for quantification. Suppl Fig.3 should become Fig. 9 as a summary. Figure 1 should be mentioned or described in the first paragraph of 2.1 Lines 71, suppl. Table 1 and suppl. Fig. 1 are from Study 1 ? Legend Fig.1, for Study 2 daily oral gavage was performed as well ? please indicate.Author Response
Reviewer 1
First of all, we thank you for your helpful and thoughtful comments on our manuscript. Thank you again for your appreciation on our study. We also greatly appreciate the comments of you, which have helped us to considerably improve manuscript. We used red font to indicate the revised portions of our manuscript. Our reply is as follows:
In Introduction, authors should provide references for efficacy, specificity and pharmacological effects of canagliflozin as an inhibitor of glucose-cotransporter.
As you requested, we added a reference for efficacy, specificity and pharmacological effects of canagliflozin as an inhibitor of SGLT to the revised manuscript (Page 4, line 20 – Page 5, line 3, and Ref# 12). In Fig. 2, did authors analyze genes of glycolysis pathway, i.e. conversion of glucose to fatty acids?
Unfortunately, we did not analyze gene expression of glycolysis in this study. This time we focused on changes of lipogenesis in the liver. Fig. 6, the brown staining of SGLT1 and 2 is diffused, appears to be cytosolic, should the staining be on plasma membrane of the hepatocytes? 
We changed and replaced to clearer IHC (Figure 6 b and d) than the original IHC in the revised manuscript. These new pictures showed that SGLT, especially SGLT2, may be stained on plasma membrane of the hepatocytes. Negative control should be shown to exclude false positives. Western blot of these proteins would be better for quantification.
We agree with you. We added negative control using human monocyte (THP-1 cell) and endothelial cell lines (HUVEC) to pictures of Western blot to supple Figure 2 in the revised manuscript. Suppl Fig.3 should become Fig. 9 as a summary.
As you requested, suppl. Figure 3 was changed to Figure 9 as a summary in the revised manuscript. Figure 1 should be mentioned or described in the first paragraph of 2.1 Lines 71.
As you requested, we mentioned Figure 1 in the first paragraph of 2.1 and of 2.2 result sections (Page 5, line 12-13, and Page 6, line 6-9). Suppl. Table 1 and suppl. Fig. 1 are from Study 1?
Yes, they are. Legend Fig.1, for Study 2 daily oral gavage was performed as well? Please indicate. Yes, it was. As you requested, we indicated this information in the revised manuscript (Page 27, line 8-9).
Reviewer 2 Report
Dear Authors
I would like to thank you for submitting your manuscript entitled “The SGLT2 inhibitor canagliflozin prevents carcinogenesis in mouse model of diabetes and non-alcoholic steatohepatitis-related hepatocarcinogenesis: Association with SGLT2 expression in hepatocellular carcinoma” to International Journal of Molecular Science. In the present study, authors demonstrated anti-cancer effect of SGLT2 inhibitor canagliflozin using NASH and hepatocellular cancer model. This aspect is very interesting and important, because cancer is the leading cause of death in Japanese patients with diabetes mellitus and liver cancer is 2-fold higher risk in patients with diabetes mellitus. I have questions and comments to authors before accept.
Major
Serum insulin level should be measured, because insulin is of the most important risk factor for NASH and liver cancer. Figure 6; IHC is not clear. This should be done again. Supplementary Fig. 2; Internal control of WB, such as GAPDH, should be shown. CCND and Cdk4 are cell cycle regulator at G1-S phase entry. However S phase was not changed in FACS analysis, why? Because CCND activity is regulated by CCND gene expression, qRT-PCR should be demonstrated. In vivo experiment, tumor proliferative activity should be examined using proliferative markers, such as Ki62.
Minor
Why Cana (5-16W) could not decrease steatohepatitis (Figure 3), even though Cana (5-16W) decreased tumorigenesis (Figure 4, 5). The mechanism by which canagliflozin decreased NASH and liver cancer are not same? This point should be discussed. STAM is very unique NASH model mice. Cangliflozin 30mg/kg/day is super pharcological dosage. These two point should be disused as experimental limitation in Discussion part.Author Response
Reviewer 2
First of all, we thank you for your helpful and thoughtful comments on our manuscript. Thank you again for your appreciation on our study. We also greatly appreciate the comments of you, which have helped us to considerably improve manuscript. We used red font to indicate the revised portions of our manuscript. Our reply is as follows:
Major:
Serum insulin level should be measured, because insulin is of the most important risk factor for NASH and liver cancer.As you requested, we measured serum insulin levels and added these values to Table 1 and supple Table 1 in the revised manuscript. Figure 6; IHC is not clear. This should be done again. As you requested, we replaced to clearer IHC (Figure 6 b and d) than the original IHC in the revised manuscript. Supplementary Fig. 2; Internal control of WB, such as GAPDH, should be shown.
As you requested, we added action as an internal control of WB to supple Fig.2 in the revised manuscript. CCND and Cdk4 are cell cycle regulator at G1-S phase entry. However S phase was not changed in FACS analysis, why?
We agree with your suggestion that S phase was supposed to be changed in our study. In actually, we found no significant reduction in percentage of S phase population in HepG2 cells treated with canagliflozin. However, although not statistically significant, the percentage of S phase population tended to be lower after treatment with canagliflozin 10 mM, compared with the control. We added the percentage of S phase population in the canagliflozin treatment and the control to the text of the revised manuscript (Page 9, line 14-16). Because CCND activity is regulated by CCND gene expression, qRT-PCR should be demonstrated.
As you requested, we evaluated gene expression of CCND1 by qRT-PCR in both HepG2 cells treated with and without canagliflozin. We added this analysis (Page 9, line 23- Page 10, line 2) to the revised manuscript as supple Fig.3. In vivo experiment, tumor proliferative activity should be examined using proliferative markers, such as Ki62.
Unfortunately, there were no liver samples left. We were sorry to inform you of this fact.
Minor:
Why Cana (5-16W) could not decrease steatohepatitis (Figure 3), even though Cana (5-16W) decreased tumorigenesis (Figure 4, 5). The mechanism by which canagliflozin decreased NASH and liver cancer are not same? This point should be discussed.
As you requested, we added the discussion about this point to the revised manuscript (Page 13, line 1-5).
STAM is very unique NASH model mice. Canagliflozin 30mg/kg/day is super pharmacological dosage. These two point should be disused as experimental limitation in Discussion part.
As you requested, we added these two limitations to the revised manuscript (Page 14, line 1-4).
Round 2
Reviewer 2 Report
Dear Authors
Thank you for your revised manuscript.
This is very important investigations.
Congratulations.
Best regards,